# The Impact of School Leadership on Inclusive Education Literacy: Examining the Sequential Mediation of Job Stress and Teacher Agency

**DOI:** 10.3390/bs15111572

**Published:** 2025-11-17

**Authors:** Yulu Feng, Dan Zhou, Yihong Wei

**Affiliations:** 1Jing Hengyi School of Education, Hangzhou Normal University, Hangzhou 311121, China; fengyulu@stu.hznu.edu.cn (Y.F.); anchor411@163.com (Y.W.); 2Zhejiang Philosophy and Social Science Laboratory for Research in Early Development and Childcare, Hangzhou Normal University, Hangzhou 311121, China

**Keywords:** inclusive education, school leadership, job stress, teacher agency, inclusive education literacy, chain mediation

## Abstract

Based on the Job Demands-Resources model theory, this study examines how school leadership affects inclusive education literacy among teachers in regular classrooms, focusing on the mediating roles of job stress and teacher agency. Using validated scales measuring school leadership, job stress, teacher agency, and inclusive education literacy, data from 751 inclusive education teachers in Sichuan, China, using a combined cluster and stratified sampling method, were analyzed via structural equation modeling. Results indicate school leadership directly enhances inclusive education literacy while also operating through job stress and teacher agency as parallel mediators. A significant chain mediation pathway further reveals a sequential stress reduction-empowerment process. We recommend that principals implement dual leadership strategies combining burden alleviation with empowerment: applying distributed leadership to optimize workflows alongside transformational leadership to foster collaboration and activate teacher agency, thereby systematically improving inclusive education quality.

## 1. Introduction

Inclusive education is evolving from expanding access to enhancing quality—a shift that focuses attention directly on classroom practice and student outcomes. In this context, teachers’ inclusive education literacy has become pivotal to successful inclusion. China’s 2022 Guidelines for Evaluating the Quality of Special Education emphasize strengthening teacher development mechanisms to improve professional competence in special education and foster teaching innovation ([22]). Similarly, the OECD’s 2023 Indicators for Inclusive Education: An Analytical Framework identifies teacher inclusive education literacy as a key quality indicator ([13]). This literacy encompasses the knowledge, skills, attitudes, and support-seeking capacities teachers demonstrate in inclusive settings ([7]). For teachers in learning in regular classrooms—the primary implementers—their competency directly determines educational effectiveness. Yet these teachers face complex demands, including disability identification, IEP implementation, and curriculum adaptation, leading to considerable job stress ([1]). Such stress arises when professional demands conflict with teachers’ perceived capacity to fulfill their roles ([23]). Many feel inadequately prepared in specialized knowledge and skills, further intensifying work pressure ([32]), which depletes psychological resources and impedes professional growth. This reality underscores the need for supportive organizational environments. School leadership significantly shapes such environments, as an interactive process between leaders and staff, it is essential for building functional schools and directly or indirectly influencing teacher practice ([24]). Leadership serves as a critical job resource with dual functions: as a buffer that reduces job stress through workflow optimization and professional support ([12]), and as a catalyst that activates teacher agency through empowerment and trust, encouraging pedagogical innovation and professional learning ([27]). Teacher agency refers to educators’ proactive, constructive responses to teaching diverse learners in general classrooms ([45])—a higher-level psychological capacity that strongly promotes inclusive education literacy when fostered in teachers. However, existing research lacks mechanistic insights into how such literacy develops, particularly among learning in regular classrooms teachers. Most studies assess specific competencies in isolation, failing to reveal the dynamic pathways through which factors interact, thereby limiting effective intervention design. To address this gap and support high-quality inclusive education, this study examines how school leadership predicts inclusive education literacy through the chain mediation of job stress and teacher agency. Clarifying this pathway offers theoretical depth to understanding leadership’s role in teacher development and provides schools and policymakers with evidence-based strategies for systemic improvement of inclusive education quality.

## 2. Literature Review and Research Hypotheses

### 2.1. School Leadership and Teacher Inclusive Education Literacy

In student-centered schools, teachers play a pivotal role in helping students realize their potential. Their continuous professional development significantly impacts student learning outcomes. Teacher literacy refers to the ability to integrate psychological resources for effective teaching in specific educational contexts ([28]). Teacher inclusive education literacy, however, represents a more integrated, high-level construct encompassing the knowledge, skills, attitudes, and support-seeking capacities teachers demonstrate in inclusive settings to ensure effective learning for all students ([46]). The Job Demands-Resources (JD-R) model analyzes occupations through job demands and resources, which jointly influence employee psychology and performance via ‘health impairment’ and ‘motivational gain’ pathways ([2]). This framework aligns well with inclusive education literacy, which requires not only solid knowledge and positive attitudes but also the practical wisdom to seek support when facing complex demands. Empirical evidence confirms the importance of resources for teachers in learning in regular classrooms. Kuyini found that general teachers consider specific teaching competencies crucial for inclusion, with human and material resources being key supports ([11], [10]). Wang also identified support-seeking ability as critical for measuring this literacy ([37]).

School leadership represents the most crucial organizational-level job resource for these teachers. Both transformational leadership, with its emphasis on vision and individual consideration ([4]), and distributed leadership, which focuses on empowerment and shared decision-making ([39]), provide essential psychological and systemic support. These leadership behaviors—offering clear direction, emotional support, professional development opportunities, and autonomy—create a rich ‘resource reservoir’. Research consistently shows school leadership significantly impacts teacher literacy ([8]). Principals directly influence teachers’ acquisition of inclusive knowledge through learning platforms ([40]) and encourage proactive skill development through empowerment ([34]). The Council of Chief State School Officers further emphasized this by issuing an Inclusive Leadership Guide with nine strategies to help principals address inclusion challenges ([5]), confirming leadership’s fundamental role in driving teachers’ professional output.

### 2.2. Job Stress and Teacher Agency

Teacher job stress arises when professional demands exceed coping capacity, creating unpleasant psychological experiences ([17]). For teachers in learning in regular classrooms, multiple high-difficulty demands—including student assessment, IEP implementation, and curriculum adaptation—directly trigger stress. Without adequate resources, chronic stress depletes psychological resources through the health impairment pathway, potentially leading to burnout and inhibited professional growth ([6]). While some studies note stress might initially drive teaching practice, excessive stress typically reduces self-efficacy and job satisfaction, creating overall negative impacts ([9]). Moderate stress levels, however, can reduce environmental pressure and enhance positive behaviors ([3]), indicating stress’s complex dual role as both driver and barrier.

Teacher self-efficacy forms the foundation of teacher agency. However, current research in this field demonstrates three primary limitations: predominant reliance on qualitative methods, narrow sociological perspectives, and excessive focus on influencing factors and motivational strategies ([41]). Ecosystem theory offers a comprehensive framework that addresses these constraints by advocating for support systems that concurrently develop teachers’ self-support mechanisms and external support structures, establishing integrated platforms for teacher agency development ([38]). This holistic approach enables teachers’ self-efficacy to be reinforced through coordinated internal and external support, thereby enhancing their sense of professional meaning. Teacher development should therefore balance individual growth with ecological support through an integrated internal-external system for stimulating agency. Conceptually, teacher agency represents a dynamic, multidimensional capacity expressed through action rather than a fixed attribute ([26]), requiring teachers’ critical consciousness, contextual adaptability, and systematic support ([14]). Within the JD-R framework, abundant job resources are crucial for activating the motivational pathway. When principals provide resources through empowerment, trust, and professional support, they significantly strengthen teachers’ autonomy, efficacy, and sense of purpose, thereby directly fostering agency. Teachers with strong agency proactively engage in pedagogical reflection, develop innovative instructional strategies, and pursue continuous professional learning, directly advancing their inclusive education literacy.

Job stress and teacher agency interact dynamically ([42]). Persistent stress depletes the psychological energy needed for agency ([20]), while higher self-efficacy improves stress adaptation. Interestingly, under specific conditions, stress might positively influence agency ([31]), and agency itself enhances stress coping capacity ([21]), suggesting a potential virtuous cycle.

In practice, however, principals often face challenges with low teacher participation willingness and professional competence when promoting inclusion ([18]). Increased workload creates significant pressure, reducing teacher proactivity. Therefore, this study examines whether school leadership can enhance inclusive education literacy through ‘stress reduction’ and ‘empowerment’ mechanisms.

### 2.3. Research Hypotheses

While no study has specifically examined how school leadership enhances inclusive education literacy among teachers in learning in regular classrooms, existing research indicates its direct predictive effect. This study surveyed 751 such teachers in Sichuan Province, aiming to: Theoretically, validate—based on the JD-R model—the internal mechanism whereby school leadership predicts inclusive education literacy through the chain mediation of ‘job stress → teacher agency’, revealing the complete ‘resources → stress reduction → empowerment → performance’ pathway and deepening understanding of teachers’ professional development processes. Practically, provide precise empirical evidence for systematically developing leadership strategies and teacher support systems that balance ‘burden reduction’ with ‘capacity empowerment’, thereby effectively advancing high-quality development of inclusive education teachers. The hypothesized relationships are shown in Figure 1. Research hypotheses are:

**H1.** 
*School leadership positively predicts teachers’ inclusive education literacy.*


**H2.** 
*Job stress mediates between school leadership and inclusive education literacy.*


**H3.** 
*Teacher agency mediates between school leadership and inclusive education literacy.*


**H4.** 
*Job stress and teacher agency form a chain-mediating pathway between school leadership and inclusive education literacy.*


## 3. Research Methods

### 3.1. Procedure

This cross-sectional study employed a quantitative research method, combining cluster sampling and stratified sampling, to conduct a survey of teachers in Sichuan Province, China. The sampling strategy accounted for regional variations in socioeconomic development and inclusive education implementation. Primary sampling units were randomly selected cities representing varying levels of socioeconomic development and inclusive education progress. Within these cities, we conducted stratified sampling of schools implementing learning in regular classrooms, selecting one school from each administrative district. All teachers involved in learning in regular classrooms at selected schools were invited to participate. Data collection utilized four validated scales. Analyses were performed using SPSS 25.0 and AMOS 22.0, including confirmatory factor analysis for scale validation, correlation analysis, common method bias testing, and structural equation modeling for hypothesis testing. The survey assessed current levels of school leadership, job stress, teacher agency, and inclusive education literacy among elementary and secondary school teachers. Statistical analyses identified key variables and their interrelationships.

### 3.2. Data Collection and Processing

Paper questionnaires were administered during in-service teacher training sessions in 2024 through partnerships with local education authorities and special education research centers. Participating schools were identified in collaboration with educational departments. Data collection occurred during scheduled teacher meetings. Each questionnaire contained a detailed informed consent form explaining research purposes, voluntary participation, and data confidentiality. Completion of questionnaires implied informed consent. After standardized instructions from trained investigators, teachers completed questionnaires independently with immediate collection. Across multiple training sessions over two months, 767 questionnaires were returned. Data quality checks excluded incomplete, inconsistent, duplicated, illegible, or patterned responses, yielding 751 valid questionnaires (97.9% validity rate). Participant demographics are presented in Table 1.

### 3.3. Measurement Tools

(1)School Leadership

The School Inclusive Climate Scale developed by Schaefer was adopted and underwent comprehensive cultural adaptation ([29]). The adaptation process included: initial revision incorporating Chinese cultural context, back-translation, evaluation by an expert committee from linguistic, cultural, and measurement perspectives, followed by small-scale pilot testing and interviews. The finalized version was empirically validated with a large sample. The revised instrument comprises 9 items across two factors: Principal Support (5 items) and Practical Engagement (4 items). Confirmatory factor analysis demonstrated excellent model fit: *χ*^2^/*df* = 4.60 (*χ*^2^ = 115.02, *df* = 25), CFI = 0.99, GFI = 0.96, AGFI = 0.93, NFI = 0.99, IFI = 0.99, TLI = 0.98, RMR = 0.02, SRMR = 0.02, and RMSEA = 0.07, meeting standard requirements. The full scale exhibited excellent reliability (α = 0.97; split-half reliability = 0.89). The Principal Support subscale showed α = 0.95 and split-half reliability = 0.84, while the Practical Engagement subscale demonstrated α = 0.95 and split-half reliability = 0.93. Focusing on school leadership, this study utilized the 5-item Principal Support subscale as a unidimensional measure. CFA confirmed good model fit: *χ*^2^/*df* = 2.600, CFI = 0.996, IFI = 0.996, GFI = 0.993, TLI = 0.992, RMR = 0.011, RMSEA = 0.046, and SRMR = 0.011. The scale demonstrated good internal consistency (α = 0.888). Items were rated on a 5-point Likert scale from 1 (strongly disagree) to 5 (strongly agree), with higher scores indicating stronger agreement.

(2)Teacher Inclusive Education Literacy

The present study employed the Teacher Professional Competence in Inclusive Education Questionnaire, developed by Wang Yan’s research team ([37]), which is compliant with local culture and background. The instrument comprises 28 items organized across four dimensions: professional attitude (items 1–8), professional knowledge (items 9–14), professional skills (items 15–22), and support-seeking skills (items 23–28). The professional knowledge dimension contains 6 items assessing understanding of fundamental principles and methods for educating students with special needs. The professional attitude dimension includes 8 items measuring beliefs about educational equity for students with disabilities. The professional skills dimension consists of 8 items evaluating competencies in collaborative teaching and interdisciplinary cooperation. The support-seeking dimension comprises 6 items examining abilities to mobilize community resources for instructional support. Confirmatory factor analysis demonstrated acceptable model fit: *χ*^2^/*df* = 3.779, CFI = 0.908, RMSEA = 0.056 (90% CI: 0.053–0.059), and SRMR = 0.051. These indices meet established standards for acceptable fit (*χ*^2^/*df* between 2.0–5.0, CFI > 0.9, SRMR and RMSEA < 0.08), indicating good alignment between the theoretical model and empirical data. Reliability analysis revealed strong internal consistency across all dimensions: professional attitude (α = 0.887), professional knowledge (α = 0.891), professional skills (α = 0.902), and support-seeking skills (α = 0.834). The full questionnaire demonstrated excellent overall reliability (α = 0.939). Items were scored on a 5-point Likert scale from 1 (strongly disagree) to 5 (strongly agree), with higher scores indicating stronger agreement and greater levels of professional competence in inclusive education.

(3)Job Stress

The Job Stress Scale developed by Li was adopted in this study ([15]). The original instrument comprised 22 items across five dimensions: workload stress (6 items), student learning stress (5 items), social and school evaluation stress (4 items), professional development stress (4 items), and student behavioral issues stress (3 items). Initial psychometric evaluation demonstrated acceptable reliability, with subscale Cronbach’s α coefficients ranging from 0.72 to 0.90 and test-retest reliability between 0.86 and 0.92. Given the Chinese teacher participants in this study, a comprehensive adaptation process was implemented: first, cultural adaptation and back-translation were conducted; next, an expert committee evaluated all versions from linguistic, cultural, and measurement perspectives; finally, small-scale pilot testing and interviews were performed to finalize the instrument. The scale’s reliability and validity were subsequently empirically validated with a large sample. Confirmatory factor analysis was conducted while preserving the original theoretical structure. The modified model demonstrated satisfactory fit indices: *χ*^2^/*df* = 4.67 (*χ*^2^ = 363.87, *df* = 78), GFI = 0.93, AGFI = 0.90, NFI = 0.94, IFI = 0.95, TLI = 0.94, CFI = 0.95, RMSEA = 0.07, SRMR = 0.04, and RMR = 0.05. The full scale exhibited high internal consistency (α = 0.93) and split-half reliability (0.79), while subscales demonstrated good internal consistency (α = 0.71–0.86) and split-half reliability (0.65–0.77). These psychometric properties confirm the instrument’s validity for stress assessment in professional educational settings. The scale employs a 5-point Likert format ranging from 1 (strongly disagree) to 5 (strongly agree), with higher scores indicating stronger agreement and greater perceived job stress.

(4)Teacher Agency

This study employed the Teacher Agency Scale developed by Zhou, which is compliant with local culture and background ([45]). The instrument contains 17 items across two factors: teaching efficacy (7 items) and constructive engagement (10 items). Factor loadings ranged from 0.61 to 0.81 for teaching efficacy and from 0.53 to 0.82 for constructive engagement. Reliability analysis demonstrated excellent psychometric properties, with factor-level Cronbach’s α coefficients between 0.90–0.93 and split-half reliability coefficients of 0.84–0.87. The full scale showed strong internal consistency (α = 0.95) and split-half reliability (r = 0.82). Confirmatory factor analysis indicated acceptable model fit: *χ*^2^/*df* = 5.14 (*χ*^2^ = 575.68, *df* = 112), CFI = 0.94, GFI = 0.91, AGFI = 0.88, NFI = 0.93, IFI = 0.94, TLI = 0.93, RMSEA = 0.08, SRMR = 0.04, and RMR = 0.04. The scale utilizes a 5-point Likert format from 1 (strongly disagree) to 5 (strongly agree), with higher scores indicating stronger agreement and greater levels of teacher agency.

## 4. Results

### 4.1. Preliminary Analysis

#### 4.1.1. Common Method Bias Test

Common method bias was examined using Harman’s single-factor test in SPSS 25.0. The results revealed ten factors with eigenvalues greater than 1, with the first factor accounting for 31.25% of the variance—below the 40% threshold, indicating no substantial common method bias. While we acknowledge this method’s limitations in sensitivity and its inability to definitively rule out method bias, the results suggest that common method variance does not significantly compromise our findings ([25]).

#### 4.1.2. Descriptive Statistics and Correlations

As presented in Table 2, all four variables demonstrated significant pairwise correlations. School leadership showed positive correlations with teacher agency (*r* = 0.38, *p* < 0.01) and inclusive education literacy (*r* = 0.30, *p* < 0.01), while correlating negatively with job stress (*r* = −0.16, *p* < 0.01). Teacher agency was positively associated with inclusive education literacy (*r* = 0.55, *p* < 0.01) and negatively with job stress (*r* = −0.21, *p* < 0.01). A significant negative correlation was also observed between job stress and inclusive education literacy (*r* = −0.28, *p* < 0.01).

#### 4.1.3. Testing and Comparison of the Chain Mediation Model

Structural equation modeling (SEM) was employed to examine the chain mediation effect of job stress and teacher agency between school leadership and inclusive education literacy. Since significant pairwise correlations were found among the four variables, model construction was deemed appropriate. A model with four latent variables was established (Figure 1), with school leadership as the independent variable, inclusive education literacy as the dependent variable, and job stress and teacher agency as mediators. The model was tested using Amos 26.0. The fit indices were: *χ*^2^/*df* = 8.083, RMR = 0.028, GFI = 0.916, IFI = 0.932, TLI = 0.908, CFI = 0.931, NFI = 0.923, and RMSEA = 0.097, indicating an acceptable but not excellent fit (Table 3). The elevated RMSEA suggests some systematic discrepancy between the theoretical model and observed data, potentially due to either incomplete capture of complex relationships by the proposed chain pathway or unexplained covariance in the sample data. However, existing literature indicates that RMSEA values below 0.1 are acceptable ([30]). As shown in Figure 1, school leadership negatively predicted job stress (*β* = −0.15, *p* < 0.001) and positively predicted both teacher agency (*β* = 0.30, *p* < 0.001) and inclusive education literacy (*β* = 0.08, *p* < 0.01). This suggests that effective leadership can alleviate perceived stress, stimulate agency, and foster literacy development. Job stress negatively predicted both teacher agency (*β* = −0.22, *p* < 0.001) and inclusive education literacy (*β* = −0.13, *p* < 0.001), indicating that lower stress levels facilitate greater agency and literacy enhancement. Teacher agency positively predicted inclusive education literacy (*β* = 0.61, *p* < 0.001), confirming its crucial role in literacy development. In summary, the model reveals a direct positive prediction of school leadership on inclusive education literacy (*β* = 0.08, *p* < 0.05). Job stress acts as a negative mediator, while teacher agency serves as a positive mediator in this relationship. Furthermore, job stress directly predicts literacy, and teacher agency mediates the relationship between job stress and literacy. Ultimately, job stress and teacher agency function as chain mediators in the effect of school leadership on inclusive education literacy.

Figure 2 shows a significant direct path (*p* < 0.05) and other highly significant paths (*p* < 0.001). Joint significance tests confirm the significant chain mediation effect of job stress and teacher agency between school leadership and inclusive education literacy. The bias-corrected nonparametric percentile bootstrap method with 2000 resamples was used to further test the chain mediation effect. As shown in Table 4, the 95% confidence intervals for all three mediation paths exclude zero, confirming: (1) the significant mediating effect of job stress, (2) the significant mediating effect of teacher agency, and (3) the significant chain mediating effect of job stress and teacher agency. The direct effect of school leadership on inclusive education literacy was 0.084, the total indirect effect was 0.237, and the total effect was 0.321. The mediation effect accounted for 73.83% of the total effect, indicating that over half of school leadership’s impact on inclusive education literacy operates through the chain mediation of job stress and teacher agency (Table 4).

## 5. Discussion

### 5.1. Direct Effect of School Leadership on Inclusive Education Literacy

The findings reveal that school leadership significantly and positively predicts teachers’ inclusive education literacy. This confirms the research results of [33] ([33]) and aligns with the findings of [19] ([19]). From the perspective of the JD-R model, this outcome clarifies that school leadership, as a core job resource, plays a direct driving role in teachers’ professional performance. In the context of inclusive education, principals directly enrich the environmental resource pool essential for teacher professional growth by building clear visions, providing platforms and resources for professional development, and fostering a supportive organizational climate. This not only facilitates teachers’ access to knowledge and skills but also deepens their professional identity and positive attitudes toward inclusive education through organizational care and value guidance. Consequently, supportive leadership behaviors themselves serve as a powerful signal of professional development, directly nurturing and enhancing teachers’ inclusive literacy.

### 5.2. Mediating Role of Job Stress

The study found that job stress plays a partial mediating role in the relationship between school leadership and teachers; inclusive education literacy, providing strong support for the ‘health impairment pathway’ in the JD-R model. This study indicates that strong school leadership can serve as a crucial buffer resource for teachers’ work stress. Through providing substantive support—including coordinating professional personnel to deliver specialized assistance and optimizing workflows—as well as psychological support encompassing teacher counseling, encouragement, and recognition of teachers’ capabilities, principals can effectively mitigate the negative impact of high job demands and reduce teachers’ perceived stress levels. When teachers’ psychological resources are no longer continuously depleted by excessive stress, they gain more abundant energy and a more positive mindset to engage in professional learning and practical innovation, thereby indirectly enhancing their inclusive literacy. With various forms of supportive measures reducing job stress, teachers can subsequently devote more attention to improving their professional competence. These findings provide additional corroborating evidence for Wang Yan’s research results ([35]).

### 5.3. Mediating Role of Teacher Agency

School leadership not only predicts teachers’ inclusive education literacy through job stress but also through teacher agency. Research indicates that teacher agency plays a partial mediating role between school leadership and inclusive education literacy, which aligns with the findings of [36] ([36]). This finding vividly illustrates the ‘motivational pathway’ in the JD-R model, revealing how job resources drive performance by stimulating intrinsic motivation. In this study, principals’ empowering behaviors significantly enhanced teachers’ sense of autonomy, efficacy, and work meaningfulness—exemplified by power-sharing in distributed leadership and intellectual stimulation in transformational leadership. These valuable psychological resources create fertile ground for the emergence and development of teacher agency. Teachers who feel trusted and empowered are more likely to transition from passive task implementers to active educational explorers, engaging more proactively in teaching reflection, pursuing professional development, and creatively solving practical challenges, thereby directly promoting the comprehensive development of their inclusive education literacy. Thus, the enhancement of literacy through leadership is largely achieved by igniting teachers’ inner drive.

### 5.4. Chain Mediation Through Job Stress and Teacher Agency

Critically, this study demonstrates a significant chain mediation through job stress and teacher agency between school leadership and inclusive education literacy. These findings align with [27] ([27]) and indirectly support [43] ([43]), demonstrating how the JD-R model’s health impairment and motivational pathways interact through a sequential ‘stress reduction-empowerment’ process. Persistent stress depletes psychological resources, potentially causing fatigue or learned helplessness that suppresses agency. School leadership serves as a crucial external buffer: by reducing job stress, it preserves mental capacity—the essential foundation for agency development. Teachers freed from excessive pressure demonstrate greater initiative and resilience. This chain pathway reveals leadership support as an integrated process: resource provision first conserves psychological resources, then enables agency cultivation. This insight advances our understanding of teacher development, emphasizing the need to concurrently address survival needs (stress reduction) and growth needs (agency activation).

Furthermore, this study explored the potential moderating effects of demographic variables—including gender, age, Years of teaching and Years as an inclusive teacher —on the established mediation model linking school leadership to outcomes through job stress and teacher agency. However, preliminary analyses did not identify significant moderating effects of these factors on the mediation pathways.

## 6. Recommendations

### 6.1. Improving Leadership Styles to Provide Active Support

Principals, as educational leaders, should provide comprehensive support for teachers. They need to develop both the capacity to navigate educational reforms and leadership styles adapted to inclusive education development, ultimately fostering high-quality teaching teams and advancing inclusive education. Specifically, principals should systematically enhance their professional competence in inclusive education through regular special education training and inter-school seminars, transitioning from ‘administrative leadership’ to ‘professional leadership’. Furthermore, as leadership style determines school development direction—with transformational leadership positively influencing teachers’ inclusive education attitudes ([16])—principals should implement a ‘dual-track leadership model’. This involves using transformational leadership to establish a clear inclusive vision and innovation incentives, while applying distributed leadership to form collaborative teams comprising special educators, psychologists, and general teachers. Clear goals, adequate support, and reward mechanisms during educational reforms are crucial for teacher development.

### 6.2. Reducing Job Stress for Teachers in Learning in Regular Classrooms

School administrators should make every effort to reduce job stress for these teachers. Rational workload allocation by principals helps diminish environmental pressure and enhance positive behaviors ([3]). Schools should establish a stress reduction support system comprising: (1) institutional measures—developing workload calculation standards through consultation with teachers and professionals, properly accounting for specialized tasks like student assessment and IEP development; (2) resource support—establishing inclusive education resource centers providing standardized assessment tools, instructional adaptation plans, and professional communication channels; (3) psychological support—integrating counseling services into teacher support systems and creating peer support groups for experience sharing and emotional assistance. This integrated approach of institutional guarantee, professional support, and psychological care forms a comprehensive stress management system that mitigates negative stress effects and promotes inclusive education literacy.

### 6.3. Fully Activating Teacher Agency

Since professional growth ultimately depends on individual engagement, teacher training should emphasize agency development rather than merely transmitting inclusive education concepts. Three primary strategies can activate teacher agency: First, conduct professional strength assessments to help teachers identify their positioning and development direction in inclusive education. Second, delegate instructional autonomy by granting teachers professional discretion in curriculum adaptation and teaching methods. Principals should adopt distributed leadership practices, willingly delegating authority ([44]) and encouraging teacher participation in decision-making regarding school inclusive education development. Third, build professional learning communities that regularly organize teaching case studies and classroom observations, while establishing ‘inclusive education innovation awards’ to recognize and disseminate innovative practices. Principals should provide collaboration platforms where teachers in regular classrooms can cooperate with special educators and other colleagues, thereby stimulating initiative and promoting professional growth.

### 6.4. Coordinating Multiple Factors to Enhance Inclusive Education Literacy

Research indicates that improving inclusive education literacy involves the synergistic effect of multiple factors rather than isolated elements. Therefore, we should leverage various factors collectively through a coordinated support network. Within schools, establish inclusive education support teams led by principals and integrated across teaching, moral education, and administrative departments. Beyond schools, actively build partnerships with universities, special education schools, and agencies to provide professional development support. Simultaneously, attention should be paid to the chain effect between job stress and teacher agency—moderating stress levels can help transform pressure into motivation, which is beneficial for reducing teacher burnout and motivating teachers to actively seek resources and support ([47]). Educational administrative departments should incorporate both stress management and agency activation into school quality evaluation systems for inclusive education, while allocating special funding to support related practices.

## 7. Limitations and Future Research

This study has several limitations. First, the regional restriction to Sichuan Province limits generalizability of the school leadership’s effects. Subsequent research should include broader regional sampling. Second, reliance solely on teacher perspectives may introduce subjectivity; future designs could incorporate multiple stakeholders (principals, parents, students) for triangulation. Finally, the leadership instrument lacked style differentiation, preventing comparison of specific leadership types’ impacts. Future studies should refine leadership dimensions to examine how particular styles influence inclusive education literacy.

## 8. Conclusions

This study, grounded in the Job Demands-Resources model, systematically demonstrates how school leadership predicts teachers’ inclusive education literacy through a sequential ‘stress reduction-empowerment’ mechanism. The findings reveal that school leadership not only directly predicts inclusive education literacy but also operates through job stress and teacher agency as parallel mediators. Crucially, we identified a significant chain mediation pathway (job stress → teacher agency), confirming that psychological burden reduction enables subsequent professional empowerment. These findings advance JD-R model applications in education by revealing how job resources foster professional development through dual pathways: blocking resource depletion while activating motivation. Practically, educational administrators should implement balanced strategies combining burden reduction with capacity building. Principals can utilize distributed leadership to optimize workflows alongside transformational leadership to activate teacher agency. Policy efforts should integrate both stress management and autonomy support into inclusive education quality frameworks. While this cross-sectional study establishes predictive relationships, future longitudinal or intervention research is needed to verify causal directions.

## Figures and Tables

**Figure 1 behavsci-15-01572-f001:**
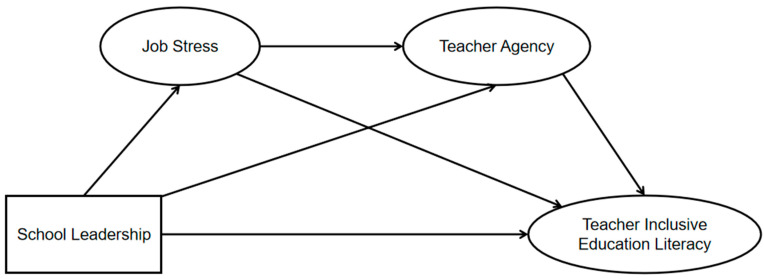
Assumed Relationship Path Diagram.

**Figure 2 behavsci-15-01572-f002:**
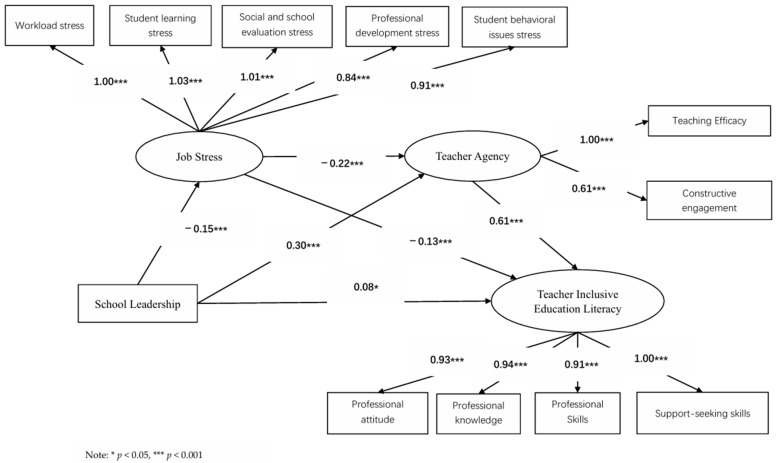
Chain Mediation Model of Job Stress and Teacher Agency in the Influence of School Leadership on Teacher Inclusive Education Literacy.

**Table 1 behavsci-15-01572-t001:** Demographics of the Participants (*N* = 751).

Characteristic		Total	Percentage
School level	Primary	680	90.5
	Junior high	71	9.5
Gender	Male	113	15
	Female	638	85
Age	21–30 years old	215	28.6
	31–40 years old	334	44.5
	41–50 years old	168	22.4
	51–60 years old	34	4.5
Years of teaching	0–2 years	84	11.2
	3–10 years	255	34
	11–20 years	412	54.9
Years as an inclusive teacher	0–2 years	323	43
	3–10 years	255	34
	11–20 years	173	23

**Table 2 behavsci-15-01572-t002:** Correlation Matrix Among School Leadership, Job stress, Teacher Agency, and Teacher Inclusive Education Literacy.

Variable Name	Mean (M)	Standard Deviation (SD)	Correlation Among Variables Teacher Proactivity			
			Teacher Inclusive Education Literacy	Job Stress	Teacher Agency	School Leadership
Teacher Inclusive Education Literacy	3.799	0.596	1			
Job Stress	3.484	0.722	−0.212 **	1		
Teacher Agency	3.449	0.718	0.546 **	−0.284 **	1	
School Leadership	4.045	0.769	0.380 **	−0.155 **	0.300 **	1

Note: ** *p* < 0.01.

**Table 3 behavsci-15-01572-t003:** Model Fit Indices for the Chain Mediation Model of Job Stress and Teacher Agency in the Influence of School Leadership on Teacher Inclusive Education Literacy.

Fitting Index	*χ* ^2^	*df*	*χ*^2^/*df*	RMR	GFI	IFI	TLI	CFI	NFI	RMSEA
Calculated Value	396.072	49	8.083	0.028	0.916	0.932	0.908	0.931	0.923	0.097
Judgment Criteria			≤5	≤0.10	≥0.90	≥0.90	≥0.90	≥0.90	≥0.90	≤0.10

**Table 4 behavsci-15-01572-t004:** Bootstrap Test Results for the Chain Mediated Effect Model.

Effects	Path	Effect Value	Proportion of the Total Effect	95% Confidence Interval	
				Lower Limit	Upper Limit
Direct Effects	Path 1: School Leadership → Teacher Inclusive Education Literacy	0.084	26.17%	0.014	0.154
Indirect Effects	Path 2: School Leadership → Job Stress → Teacher Inclusive Education Literacy	0.022	6.85%	0.008	0.042
Indirect Effects	Path 3: School Leadership → Teacher Agency → Teacher Inclusive Education Literacy	0.193	60.13%	0.135	0.255
Indirect Effects	Path 4: School Leadership → Job stress → Teacher Agency → Teacher Inclusive Education Literacy	0.022	6.85%	0.011	0.037
Total Effect Value	0.284	100%			

## Data Availability

The datasets used and analyzed during the current study available from the corresponding author on reasonable request.

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
