# Peer review of "The Impact of School Leadership on Inclusive Education Literacy: Examining the Sequential Mediation of Job Stress and Teacher Agency"

_behavsci, 2025, doi:10.3390/bs15111572_

Round 1

Reviewer 1 Report

Comments and Suggestions for Authors

The paper addresses a very interesting and relevant topic facing schools today. It's a global situation that affects most countries. I think the article is well written, although at times I find the use of headings excessive. For example, within the section on participants, there are several subsections, which I consider excessive. To streamline the reading, I propose fewer subsections. I think the practical implications at the end of the paper are very right.

Reviewer 2 Report

Comments and Suggestions for Authors

Dear Authors,

Thank you for the opportunity to review your article. Below, I provide constructive feedback to help enhance its clarity, depth, and scholarly impact. I hope these suggestions assist you in refining your work and strengthening its contributions.

Title

The Impact of School Leadership on Inclusive Education Literacy: Examining the Sequential Mediation of Job Stress and Teacher Agency

Abstract

1 – Lack of theoretical grounding in the abstract: The opening sentence is generic and disconnected from the paper’s theoretical basis (JD-R model, leadership theories, or teacher agency).

2 – Methodological vagueness: The method is mentioned superficially (“a survey of 751 teachers”) with no design, measurement, or analysis context.

3 – The study objective is not stated clearly.

4 – Results oversimplified: The abstract lists relationships but lacks quantitative indicators or context (e.g., “leadership positively predicts literacy” — how strongly?).

5 – Conclusion abrupt (lines 16–18): Recommendations read like bullet points rather than a concluding synthesis of the findings. Suggest rewriting this as a cohesive concluding statement.

Introduction

1 – The first paragraph discusses inclusive education evolution and policy but doesn’t connect directly to school leadership or teacher outcomes. Recommend tightening the global context to focus sooner on the leadership–teacher–literacy nexus.

2 – Objective lacks clarity and placement (line 78): The purpose is embedded in dense justification. Recommend converting it to a single explicit sentence.

3 – Inconsistency in construct terminology: Alternates between “inclusive literacy,” “inclusive education literacy,” and “teacher inclusion literacy.” Recommend consistent terminology.

4 – Lines 79–82: The introduction mentions the study “holds significant value,” but does not state why (e.g., theoretical contribution or practical implication). Suggest specifying the expected contribution.

Methodology

1 – No clear study design section: This section should explicitly state that this was a quantitative, cross-sectional, correlational survey design. Without this, readers can’t easily interpret the analytical approach.

2 – Sampling strategy vague: “Combination of random and cluster sampling” is too general. Clarify clusters (schools, districts?), randomization process, and how the sampling ensured representativeness.

3 – Missing contextual details of data collection: When and where were data collected? Was it before or after any key policy change? How long did it take? Was participation voluntary and anonymous?

4 – Procedures section too short (lines 314–318): Merely lists statistical tests; it should describe how the research was conducted (sequence, administration, consent process, etc.).

5 – Line 351: Instrument adaptation clarity missing – For the Job Stress Scale, authors mention “revised and adapted” but do not explain what modifications were made or how validity was re-established post-adaptation.

6 – Lack of translation/back-translation or cultural validation details: All instruments are adapted for Chinese teachers, but no mention of translation method, pilot testing, or cultural adaptation procedures.

Results

1 – Model fit index (RMSEA = 0.097) exceeds accepted thresholds: The authors call this a “good fit,” but RMSEA > 0.08 indicates only marginal or poor fit. They should acknowledge this limitation and discuss possible reasons (e.g., sample size, model complexity).

2 – Table 3: Path coefficients for “Direct Effects” are negative (−0.131) although the text says School Leadership positively predicts Literacy (β = 0.08). This is inconsistent and needs correction.

Discussion

1 – Discussion repeats results rather than interpreting them: Much of the text paraphrases numerical findings already reported in the Results. The authors restate the pattern of relationships without deeper explanation of why these relationships occur, missing theoretical insight.

2 – Theoretical integration is weak and inconsistent: Mentions the JD-R model and Bandura’s theory only briefly, without explaining how the findings extend or challenge these frameworks. Suggest elaborating how leadership acts as a “job resource” and how agency functions as a personal resource within JD-R.

3 – Phrases like “School leadership exerts an effect” or “influences” imply causality, but the Methods section only described a cross-sectional survey. Recommend rephrasing to “is associated with” or “predicts,” acknowledging the correlational nature.

4 – Results included detailed demographics, but the discussion ignores how gender, age, or teaching experience might shape these relationships. Suggest adding brief commentary.

Conclusion

1 – The conclusion restates statistical relationships already described in Results (positive/negative/mediating effects) without offering synthesis or meaning. Suggest highlighting what these findings mean for leadership practice and inclusive education theory.

2 – Should briefly state how principals or policymakers can use these findings to improve inclusive education practices.

3 – The section ends abruptly with no summative reflection.

Reviewer 3 Report

Comments and Suggestions for Authors

The manuscript addresses a timely and relevant topic within behavioral and educational psychology and fits the journal’s scope by exploring behavioral mechanisms underlying professional performance and wellbeing in educational contexts.

However, the paper would benefit from substantial revisions to improve conceptual precision, methodological transparency, and stylistic clarity. 

  • The literature review is rich but overly descriptive and repetitive. Authors should consolidate frameworks and choose one dominant explanatory model (e.g., JD-R combined with agency theory). It needs to clarify conceptual distinctions among competence, professional literacy, and inclusive education literacy.
  • The methods section should provide fuller detail on sampling procedures, instrument adaptation and validity evidence, acknowledge the limits of using only Harman’s single-factor test for common method bias, and discuss the moderate SEM model fit.
  • The discussion should move beyond restating results to interpret them through the JD-R model and intrinsic motivation theory.
  • Improve English expression, concision, and consistency.

Round 2

Reviewer 2 Report

Comments and Suggestions for Authors

Dear Authors,

Thank you for submitting the revised version of your manuscript. After a careful review, I am pleased to note that the majority of the suggestions raised in the previous round have been addressed clearly and effectively. The manuscript now shows substantial improvements in its theoretical grounding, clarity of objectives, methodological transparency, conceptual consistency, and interpretation of the findings. The discussion also engages more robustly with the JD-R framework and now offers stronger theoretical and practical implications.

Key issues previously highlighted—such as clarification of the study design, details of instrument adaptation, interpretation of model fit indices, and correction of inconsistent coefficients—have been appropriately resolved. While there may still be scope for further refinement in future work, the revisions made at this stage are sufficient and significantly enhance the overall quality and clarity of the manuscript.

I therefore consider the revisions satisfactory and accept the changes made.

Kind regards

Comments on the Quality of English Language

Dear Authors,

Thank you for submitting the revised version of your manuscript. After a careful review, I am pleased to note that the majority of the suggestions raised in the previous round have been addressed clearly and effectively. The manuscript now shows substantial improvements in its theoretical grounding, clarity of objectives, methodological transparency, conceptual consistency, and interpretation of the findings. The discussion also engages more robustly with the JD-R framework and now offers stronger theoretical and practical implications.

Key issues previously highlighted—such as clarification of the study design, details of instrument adaptation, interpretation of model fit indices, and correction of inconsistent coefficients—have been appropriately resolved. While there may still be scope for further refinement in future work, the revisions made at this stage are sufficient and significantly enhance the overall quality and clarity of the manuscript.

I therefore consider the revisions satisfactory and accept the changes made.

Kind regards